# Evidence for hidden leprosy in a high leprosy-endemic setting, Eastern Ethiopia: The application of active case-finding and contact screening

**Kedir Urgesa**[1]*, **Kidist Bobosha**[2], **Berhanu Seyoum**[2], **Fitsum Weldegebreal**[1], **Adane Mihret**[2], **Rawleigh Howe**[2], **Biftu Geda**[3], **Mirgissa Kaba**[4], **Abraham Aseffa**[2]

1 College of Health and Medical Sciences, Haramaya University, Harar, Ethiopia, 2 Armauer Hansen Research Institute, Addis Ababa, Ethiopia, 3 Department of Nursing, Madda Walabu University, Shashamene, Ethiopia, 4 School of Public Health, Addis Ababa University, Addis Ababa, Ethiopia

* bofekedir@gmail.com

**Data Availability Statement:** The authors confirm that all data underlying the findings are fully

## Abstract

Leprosy or Hansen's disease is a disabling infectious disease caused by *Mycobacterium leprae*. Reliance on the self-presentation of patients to the health services results in many numbers of leprosy cases remaining hidden in the community, which in turn results in a longer delay of presentation and therefore leading to more patients with disabilities. Although studies in Ethiopia show pockets of endemic leprosy, the extent of hidden leprosy in such pockets remains unexplored. This study determined the magnitude of hidden leprosy among the general population in Fedis District, eastern Ethiopia. A community-based cross-sectional study was conducted in six randomly selected leprosy-endemic villages in 2019. Health extension workers identified study participants from the selected villages through active case findings and household contact screening. All consenting individuals were enrolled and underwent a standardized physical examination for diagnosis of leprosy. Overall, 262 individuals (214 with skin lesions suspected for leprosy and 48 household contacts of newly diagnosed leprosy cases) were identified for confirmatory investigation. The slit skin smear technique was employed to perform a bacteriological examination. Data on socio-demographic characteristics and clinical profiles were obtained through a structured questionnaire. Descriptive statistics and binary logistic regression were used to assess the association between the outcome variable and predictor variables, and the P-value was set at 0.05. From the 268 individuals identified in the survey, 6 declined consent and 262 (97.8%) were investigated for leprosy. Fifteen cases were confirmed as leprosy, giving a detection rate of 5.7% (95%, CI: 3%, 9%). The prevalence of hidden leprosy cases was 9.3 per 10,000 of the population (15/16107). The majority (93.3%) of the cases were of the multi-bacillary type, and three cases were under 15 years of age. Three cases presented with grade II disability at initial diagnosis. The extent of hidden leprosy was not statistically different based on their sex and contact history difference (p > 0.05). High numbers of leprosy cases were hidden in the community. Active cases findings, and contact screening

available without restriction. All relevant data are within the manuscript.

**Funding:** The authors received no specific funding for this work.

**Competing interests:** The authors have no competing interests.

strategies, play an important role in discovering hidden leprosy. Therefore, targeting all populations living in leprosy pocket areas is required for achieving the leprosy elimination target.

## Author summary

Leprosy, also called Hansen's disease, is a neglected infectious disease leading to deformity and disability. Late presentation and hidden cases are the major risks of leprosy-associated disability. Although leprosy endemic pocket areas and grade II disability with a high proportion were reported in Ethiopia, studies on the burden of hidden leprosy cases are limited. Therefore, this study determined the extent of hidden leprosy cases among the general population in leprosy endemic settings in eastern Ethiopia through active case findings and contact tracing. In this community-based survey, leprosy-suspected individuals in the general population and household contacts of newly diagnosed patients with leprosy were included. Health extension workers, community-based health workers in Ethiopia, visited 16107 individuals in the selected villages and 214 leprosy suspects were enrolled in the study based on the clinical signs of leprosy suspects. Leprosy experts examined all leprosy suspects clinically and a skin slit sample was taken for bacteriological examination. After the confirmation of new cases, 48 of their households' contacts were then examined by leprosy experts. Of 262 suspects and household contacts evaluated for leprosy, 15 hidden cases confirmed, giving an overall prevalence of 9.3 per 10, 000 population. Most of them were Multi-bacillary (MB) type, and one-fourth of them were younger than 15 years of age, and three cases presented with grade II disability. Hidden leprosy was not statistically associated with participants' sex, age category, and contact history.

## Introduction

Leprosy or Hansen's disease is a disabling infectious disease caused by *Mycobacterium leprae* [1]. It is one of the neglected tropical diseases of public health importance [1,2]. Leprosy is endemic in poor countries where detection rates remain low despite availability of effective treatment [3]. Though there have been reductions of about 90% in the prevalence rate, transmission continues and remains a public health issue [4].

The global target to eliminate leprosy, a reduction in prevalence to <1 case per 10,000 population, was achieved in 2000[5] and the World Health Organization (WHO) had set a target to interrupt the transmission of leprosy globally by 2020 [6]. Leprosy nevertheless continues to be a public health problem in different parts of the world [7], with more than 200,000 new cases reported every year [8].

Although Ethiopia achieved the elimination target in 1999, it still has the second-highest disease burden in terms of leprosy in Sub-Saharan Africa (SSA) [9]. Between 2013 and 2015, 3,500 to 4,000 new leprosy cases were reported to the national tuberculosis and leprosy control program [10]. In 2019, the country reported 3,201 new leprosy patients, of whom 12.8% presented with a grade II disability, as reported by WHO [8]. Studies in Ethiopia also evidenced the persistent prevalence of childhood leprosy and disabilities with multibacillary (MB) cases in rural southern Ethiopia [11,12], which suggested the ongoing transmissions of the disease in the country [13].

Active case-findings strategies are essential for discovering hidden leprosy and are an important epidemiological tool to minimize cases, reduce incidences of disability due to leprosy, and reduce the transmission of *M. leprae* [14,15]. Moreover, the global leprosy strategy

(2016–2020) promotes early case detection by the application of active case-finding and contact management in areas of higher endemicity [13].

However, in Ethiopia cases of leprosy are detected by examining patients attending health facilities (passive case detection)[16]. This passive case detection or self-reporting of patients in the integrated leprosy-control program results in increased hidden and undiagnosed leprosy cases in the community, leading to more deformities and disability [17].

While studies in Ethiopia revealed endemic leprosy pockets [18], the extent of hidden leprosy cases is rarely addressed [9]. Therefore, this study determined the magnitude of hidden and undiagnosed leprosy using house-to-house visits as a tool for active case detection and to evaluate the household contacts of leprosy in selected leprosy endemic districts in eastern Ethiopia in 2019.

## Methods

### Ethical consideration

This study was conducted according to the Helsinki Declaration and Ethiopian research regulations. The Institutional Health Research Ethics Review Committee (IHRERC) of the College of Health and Medical Sciences, Haramaya University, Ethiopia (ref no: IHRERC/152/2018) and the Armauer Hansen Research Institute Ethics Committee (ref no: P002/18 AHRI/ERC) approved the protocol. The coordinator informed all participants in advance about the purpose and time of the survey. Participants were given information on the objectives of the study, and informed consent was obtained in writing or by thumbprint. For those participants below 18 years of age, informed consent was obtained from a parent or legal guardian. To minimize the stigma, privacy was a priority during the examination of study participants. Participants participated voluntarily and withdraw from the study at any time without any consequences. Anonymity was ensured by only having participant identification numbers included during data collection.

### Study setting, design and period

A community-based observational study was conducted in six leprosy endemic villages in the Fedis District between 5 July and 30 October 2019. Fedis is one of the high leprosy endemic districts located in East Hararghe Zone, Oromia Regional State, Eastern Ethiopia. It is located at 534 km East of Addis Ababa and 24km to the south of Harar (Fig 1).The district contains 19 rural and 2 urban villages with a total population estimated to be 133,382 persons. According to the zonal health office report, 57 and 47 new leprosy cases were receiving treatment in 2017 and 2018, resulting in a prevalence of 3.5 to 4.3 per 10,000 population (East Hararghe Zonal Health office, 2017 and 2018) (Zonal TB/Leprosy focal person communication).

### Study population, and sampling

Since leprosy occurs in clusters, one large sample from a single area would not have been a reliable estimate of leprosy. Estimating the disease burden by conventional sampling procedure is difficult due to the large sample size requirement. Therefore, inverse sampling procedure was used [19,20]. Fedis District was selected among 12 high leprosy endemic districts in the East Hararghe Zone. From the District, six villages with a leprosy endemicity burden, with a total population of 35,673, were included randomly. All suspected cases and consenting individuals were screened through house-to-house visits and consecutively enrolled for leprosy diagnosis. Study criteria excluded those on multi-drug therapy at the initiation of the study and a person who lived for less than six months in the selected villages.

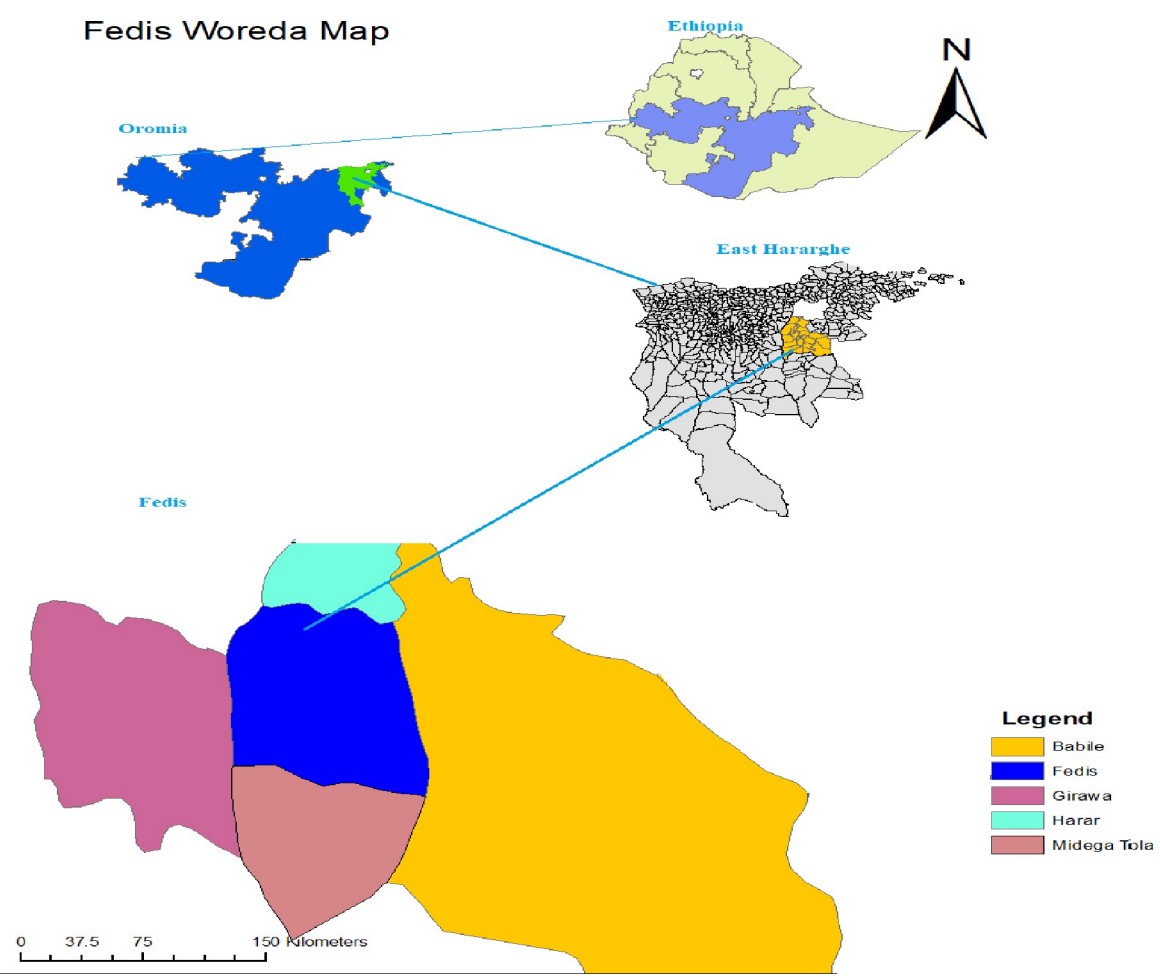

**Fig 1. Map showing the location of study site in Ethiopia 2019: Fedis district, drawn using ARCGIS version 10.1.** (source: "Natural Earth. (http://www.naturalearthdata.com/about/terms-of-use/).

### Data collection procedure and tools

**Survey.**   Full village surveys of the selected villages were conducted, including all household members. Twelve trained health extension workers (HEWs) conducted the house-to-house visits to identify leprosy suspects. Using checklists, which was adopted from the national guideline, HEWs identified suspects by showing color photos of leprosy cases and asking if any household members had similar symptoms. Suspects were referred to the nearby health facility and examined by leprosy experts from Armauer Hansen Research Institute and the research team (researchers, health officers, and HEWs).

**Physical examination.**   All individuals suspected of having leprosy underwent a standardized physical examination, as recommended by WHO and the national guidelines [21]. Briefly, the physical examination focused on examination of the skin from head to toe, including the front and back sides, the presence of skin lesions (patches or nodules), loss of sensation over the skin lesions (patches) using a "wisp of cotton wool", and the number of skin lesions counted, if any. Palpation of the nerves was checked for cord enlargement and/or tenderness, and examination of eyes, hands and feet for any disabilities[22].

**Bacteriological examination.**   According to the national guidelines, the slit skin smear examination was performed for questionable cases to confirm the diagnosis; and was also used

for leprosy classification. One slide, with smears taken from two sites (ear lobes and active lesion), was collected for examination and evaluation for *M. leprae* (acid-fast bacilli)[16]. Accordingly, the principal investigators obtained skin smears for bacteriological examination. Briefly, slit-skin smears were taken from ear lobes and skin lesions from 43 study participants. The slit-skin smears made on the slide were stained by the Ziehl-Neelsen technique, using 1% carbol fuchsin, 1% acid-alcohol and 0.25% methyl blue. Under oil immersion objective, red acid-fast bacilli were observed, arranged singly or in groups (cigar-like bundles), and bound together by a lipid-like substance, forming glia. The criteria used for diagnosis and classification were based on the local leprosy control program and followed WHO guidelines as either paucibacillary (PB) or multibacillary (MB) type[21].

After confirmation of the leprosy diagnosis, the leprosy experts determined the degree of disability and initiated multi-drug therapy. The household contacts were then scheduled for screening by the research team. Leprosy experts or dermatologists then examined the household contacts. Suspects with other skin diseases were linked to the nearby health center.

**Questionnaire.** Structured questionnaires were administered to suspected cases and household contacts to obtain data on demographic characteristics and clinical history. Information related to leprosy diagnosis was obtained, including WHO classification of leprosy, disability grade, the clinical profile of individuals including BCG scar, contact history of leprosy, and any previous history of leprosy was documented.

**Quality control.** Health extension workers were trained in the clinical examination for leprosy diagnosis and how to refer suspected cases to the nearby health center for further investigation by leprosy experts. HEWs conducted an interview in the local language (Afan Oromo) and checklists were completed by face-to-face interviews for recruitment.

## Study variables

The outcome variable was the magnitude of the hidden leprosy case. The independent variables include age, sex, occupation, residence, educational status, marital status, BCG scar, and contact history of a patient with leprosy.

## Operational definition

Suspect is an individual who presented with pale or reddish patches (skin patch with discoloration) on the skin, painless swelling or lumps in the face and earlobes, loss of, or decreased sensation on the skin, numbness or tingling of the hands and/or feet, weakness of eyelids, hands or feet, painful and/or tender nerves, burning sensation in the skin or painless wounds or burns on the hands or feet [16].

A leprosy case is a person with one of the cardinal signs of leprosy, and who requires chemotherapy. The cardinal signs of leprosy are ONE of the following: hypo-pigmented skin lesion with definite loss of sensation, thickened (enlarged) peripheral nerve with or without tenderness, and/or the presence of acid-fast bacilli in a slit-skin smear [16].

The PB type is a patient who is skin smear negative and/or the number of skin lesions is 1–5 without demonstrated presence of bacilli in the smear [23].

The MB type is a patient who is skin smear positive and/or the number of skin lesions is more than five, with demonstrated presence of bacilli in the smear, irrespective of the number of skin lesions [23].

Physical disability in leprosy is defined by the WHO in three categories [24]: Grade 0: the absence of disability (no anesthesia) and no visible damage or deformities of eyes, hands and feet; Grade I disability: the loss of protective sensibility in the eyes, hands or feet, but no visible damage or deformities; and Grade II: the presence of deformities or visible damage to the eyes

(lagophthalmos and/or ectropion, trichiasis, corneal opacity, difficulty counting fingers at 6 meters), visible damage on hands or feet (hand with ulcerations and/or traumatic, resorption, claw, fallen hand, ulcers; feet with trophic and/or traumatic injuries, resorption, claw, foot drop, ulcers, ankle contracture) [25].

Household contact is a family member or any person that who lived under the same roof with the index case for more than six months [26]. Co-prevalent leprosy is where the contacts diagnosed with leprosy at the first examination after the index case were diagnosed [27].

### Data management and statistical analysis

Data were entered in Epi-Data version 3.1 and analyzed using STATA version 13.0. Descriptive statistics such as mean and percentages were used to describe the socio-demographic characteristics and the magnitude of hidden leprosy cases. Descriptive statistics such as mean and proportion and binary logistic regression analysis were used to assess the association between the dependent and predictor variables. The significant association was declared at p-value $< 0.05$.

## Results

### Demographic characteristics of the study participants

The HEW visited 16,107 individuals during a house-to-house survey and household contact (HHC) tracing. Of these, 268 were eligible, 262 (97.8%) of whom consented to participation and were enrolled in the study. Among the volunteers who were evaluated for leprosy, 214 participants were identified as suspects for leprosy during the house-to-house visit, and 48 were household contacts of newly diagnosed cases. The mean (+ SD) age of the participants was 26.9 (±15.2) years. About 45% of the participants were female and 62% were rural residents. About half (48%) of the participants had no formal education (Table 1).

### Prevalence of hidden leprosy

During active case-finding through the house-to-house visits, 214 suspects were evaluated both clinically and/or bacteriologically (Fig 2). Thirty (14%) of the suspects had histories of contact with treated leprosy patients. Of the 214 suspects, 11 leprosy cases were confirmed, giving a detection rate of 5.1% (95%, CI = 2%, 9%). Among the newly confirmed leprosy cases, one patient had a prior history of leprosy (relapse case) and three cases had a contact history with a treated leprosy patient. The majority (90.9%) of cases were MB type leprosy, and two of them presented with grade II disability. Most (63.6%) of cases were farmers and 81.8% were male.

Following the confirmation of 11 new cases, leprosy experts and dermatologists examined 48 HHCs through contact management strategy. Among the 48 HHCs, four new leprosy cases (co-prevalent cases) were confirmed, giving an 8.3% detection rate (95%, CI = 2%, 19%). Among the co-prevalent cases, all of them were MB and two cases were under 15 years of age.

By considering both suspects and HHCs evaluations, 15 participants were found to be leprosy cases, giving a detection rate of 5.7% (95%, CI: 3%, 9%). This yields a total population-based prevalence of hidden leprosy to be 9.3 per 10,000 population. The majority 14(93.3%) of the newly diagnosed hidden cases were MB, and three cases demonstrated grade II disability. Among the newly diagnosed hidden cases, three were under 15 years of age and about one-fourth were female.

The extent of hidden leprosy was not statistically different based on their age category and contact history difference (p > 0.05). Furthermore, in the binary logistic regression analysis,

**Table 1. Distribution of demographic and clinical condition among participants with or without leprosy in Fedis District, 2019(n = 262).**

| Variables | Hidden leprosy | | | | | |
|---|---|---|---|---|---|---|
| | Negative | | Positive | | Total | |
| **Sex** | Number | % | Number | (%) | Number | % |
| Male | 134 | 92.4 | 11 | 7.6 | 145 | 55.3 |
| Female | 113 | 96.6 | 4 | 3.4 | 117 | 44.7 |
| **Total** | | | | | 262 | 100 |
| **Age category** (in years) | | | | | | |
| < 15 | 60 | 95.2 | 3 | 4.8 | 63 | 24.1 |
| 15–30 | 99 | 94.3 | 6 | 5.7 | 105 | 40.1 |
| 31–45 | 66 | 97.1 | 2 | 2.9 | 68 | 25.9 |
| Above 45 | 22 | 84.6 | 4 | 15.4 | 26 | 9.9 |
| **Total** | | | | | 262 | 100 |
| **Educational status** | | | | | | |
| No formal education | 122 | 96.8 | 4 | 3.2 | 126 | 48.1 |
| Literate | 125 | 91.9 | 11 | 8.1 | 136 | 51.9 |
| **Total** | | | | | 262 | 100 |
| **Marital status** | | | | | | |
| Single | 93 | 92.1 | 8 | 7.9 | 101 | 38.6 |
| Married | 154 | 95.6 | 7 | 4.4 | 161 | 61.4 |
| **Total** | | | | | 262 | 100 |
| **Occupation** | | | | | | |
| Farmer | 139 | 95.2 | 7 | 4.8 | 146 | 55.7 |
| Employed | 15 | 100.0 | 0 | 0.0 | 15 | 5.7 |
| Unpaid | 93 | 92.1 | 8 | 7.9 | 101 | 38.6 |
| **Total** | | | | | 262 | 100 |
| **Residence** | | | | | | |
| Rural | 155 | 95.7 | 7 | 4.3 | 162 | 61.8 |
| Urban | 92 | 92.0 | 8 | 8.0 | 100 | 38.2 |
| **Total** | | | | | 262 | 100 |

the detection rate of hidden leprosy cases was not statistically different based on their sex difference (P>0.05) (Table 2).

## Discussion

This study revealed the high prevalence of hidden leprosy in the general population. All co-prevalent patients were detected without having significant neuronal or visible physical

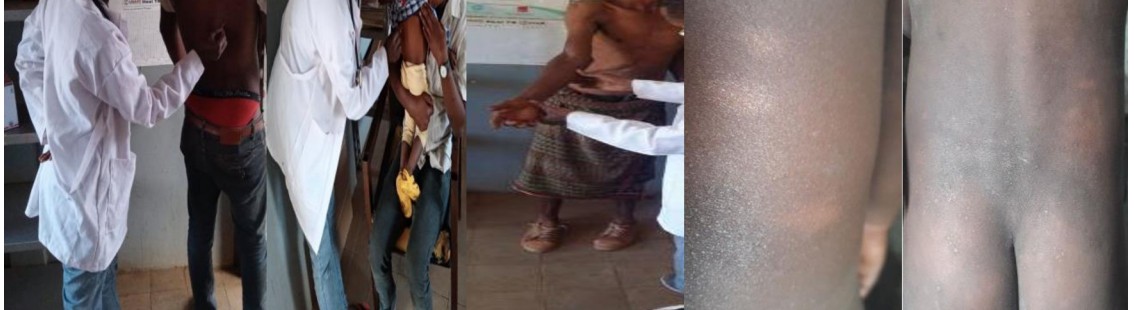

**Fig 2. Physical examination of individuals suspected for leprosy based on cardinal signs at leprosy clinic, Fedis district, 2019.**

**Table 2. Association between dependent and predictor variables among study participants in leprosy endemic setting, Fedis District, 2019(n = 262).**

| Variables | | OR | [95%Conf. Interval] | | p-value | AOR* | [95%Conf. Interval] | | p-value |
|---|---|---|---|---|---|---|---|---|---|
| **Sex** | Male | 1 | | | | | | | |
| | Female | 0.43 | 0.13 | 1.39 | 0.15 | 0.45 | 0.13 | 1.47 | 0.18 |
| **Marital status** | Single | 1 | | | | | | | |
| | Married | 0.52 | 0.18 | 1.50 | 0.23 | 0.63 | 0.17 | 2.30 | 0.48 |
| **Educational Status** | No formal education | 1 | | | | | | | |
| | literate | 2.68 | 0.83 | 8.65 | 0.09 | 1.73 | 0.41 | 7.29 | 0.45 |
| **Residence** | Rural | | | | | | | | |
| | Urban | 1.92 | 0.67 | 5.48 | 0.22 | 1.72 | 0.55 | 5.35 | 0.34 |

AOR* = adjusted odds ratio, variables in the final model

damage at the initial stage of screening. Hidden leprosy was not associated with participants' demographic characteristics and contact histories. Therefore, the presence of pockets of high endemicity with a high prevalence rate of 9.3 per 10,000 population points to the arduous journey ahead for leprosy elimination in Ethiopia.

Ethiopia, with the introduction of multi-drug treatment (MDT), achieved the elimination target at the national level with a record of 0.3 per 10,000 population in 2018 [22]. Our finding is higher than the national prevalence and that of Gambella regional state, which was 2.4 per 10,000 in 2016 [22]. We used an active case detection strategy, compared to the above-mentioned lower estimates, which used passive case detection. However, the national leprosy control program recommended the voluntary self-reporting (passive case detection) strategy. Moreover, the higher prevalence is evidence for the poor performance of passive case detection compared with active case findings [28] and active case finding is an important epidemiological tool to minimize hidden leprosy cases [15].

In this study all co-prevalent patients among HHCs were detected without having a significant neuronal or visible physical damage at the initial stage of screening. This is an indication of the feasibility and contribution of active case-finding programs to promote early case detection by tracking HHCs [13,29]. We found that one in five cases presented with grade II disability on diagnosis, showing a prolonged delay in health-seeking. This is in harmony with the research conducted in Addis Ababa, where the proportion of grade II disability among new leprosy cases was 23.7% [30]. This finding is higher than the national report of 13.6% in 2016 [22]. The higher proportion of grade II disability in the current study supports the late case presentation and ongoing transmission of leprosy [26,31]. It also reflects inadequate monitoring in the national leprosy control program [13] and the ongoing transmission of leprosy has not been interrupted [26]. Unfavorable attitude toward leprosy among the community in the same study setting [32] contributes to late presentation [33].

The proportion of childhood prevalence (20%) in this study is higher than the national prevalence (11.7%) and that of Oromia regional state (13.3%) [22]. The presence of childhood leprosy among new cases suggested the existence of the active source of infection [34] and high ongoing transmission of the disease in the community [22]. The higher proportion of childhood leprosy also a late performance indicator of the national leprosy control program [35].

This study revealed that hidden leprosy is not significantly associated with participants' contact history with leprosy and their sex difference. Similar findings have been reported in other countries [15,36]. All study participants resided in shared vulnerable areas with high leprosy endemicity villages and the same environmental exposure status [37,38]. Likewise, more

than half of the community in Fedis District was food insecure [39]. Food shortage is shown as an important poverty-related predictor of the clinical manifestation of leprosy and the greatest risk [40]. Therefore, they have the greatest risk of leprosy [41]. Hence, a higher prevalence of leprosy is expected in this district. Also, the unfavorable attitude in the general community in Fedis District and the stigma favors the hiding of patients from the diagnosis, irrespective of their sex and contact history [32,42].

### Strength and limitation of the study

This community-based active survey evidences the hidden leprosy cases that were missed by passive case detection in an endemic-leprosy setting. This study discovers leprosy patients who didn't seek health care before the inclusion. These leprosy patients are hidden within the general population and risk for themselves and others. All examinations of suspects were done in accordance with the national guidelines for leprosy diagnosis. Experienced leprosy experts and dermatologists performed clinical examinations. Learning from the successes of other disease prevention and improved health service utilization or health-care seeking through the deployment of health extension program in Ethiopia[43–45], we used the trained health extension workers as data collectors to discover hidden leprosy. Using the existing health extension programs in a context of limited resources is more workable and provides more reliable data.

The inclusion of suspects was based on questioning individuals according to the leprosy symptoms; individuals cannot recognize painless symptoms or do not report to the HEW during the house-to-house visits due to fear of stigma [33,46](selection bias). However, the colored picture used during the interview helped in recognizing symptoms.

## Conclusions and recommendations

### Conclusions

The overall prevalence of hidden leprosy is higher than the national and regional figures. An active finding and tracing of HHC in regions where leprosy is highly prevalent, like Fedis District, is an important strategy to promote early diagnosis, minimize hidden leprosy and prevent severe outcomes. The prevalence of hidden leprosy was not significantly different based on the contact history and demographic characteristics of the participants.

### Recommendations

An outreach activity of active case-finding targeting all age and sex group populations in leprosy pocket areas is crucial to stop leprosy and its complications.

It is important to develop a framework that incorporates leprosy case-finding and HHC tracing strategies in the implementation of the health extension program.

Further studies considering larger sample size and different study design need to be undertaken to identify potential factors associated with hidden leprosy.

## Acknowledgments

The authors would like to acknowledge the study participants, the district health office and data collectors for Haramaya University and AHRI their support in the study. Our great appreciation also goes to project supervisors and colleagues for their support.

## Author Contributions

**Conceptualization:** Kedir Urgesa, Abraham Aseffa.

**Data curation:** Kedir Urgesa, Kidist Bobosha, Berhanu Seyoum, Fitsum Weldegebreal, Biftu Geda, Mirgissa Kaba.

**Formal analysis:** Kedir Urgesa, Kidist Bobosha, Berhanu Seyoum, Biftu Geda, Mirgissa Kaba.

**Funding acquisition:** Kedir Urgesa.

**Investigation:** Kedir Urgesa.

**Methodology:** Kedir Urgesa, Kidist Bobosha, Berhanu Seyoum, Biftu Geda, Mirgissa Kaba, Abraham Aseffa.

**Project administration:** Kidist Bobosha, Berhanu Seyoum, Abraham Aseffa.

**Resources:** Kidist Bobosha, Berhanu Seyoum, Fitsum Weldegebreal.

**Software:** Biftu Geda.

**Supervision:** Kidist Bobosha, Berhanu Seyoum, Fitsum Weldegebreal, Adane Mihret, Rawleigh Howe, Biftu Geda, Abraham Aseffa.

**Validation:** Kedir Urgesa, Biftu Geda, Mirgissa Kaba.

**Visualization:** Kedir Urgesa, Kidist Bobosha, Berhanu Seyoum, Biftu Geda, Abraham Aseffa.

**Writing – original draft:** Kedir Urgesa, Kidist Bobosha, Berhanu Seyoum, Adane Mihret, Rawleigh Howe, Abraham Aseffa.

**Writing – review & editing:** Kedir Urgesa, Kidist Bobosha, Berhanu Seyoum, Fitsum Weldegebreal, Biftu Geda, Mirgissa Kaba, Abraham Aseffa.

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
