## [Decision Letter · Decision Letter 0]

8 Mar 2021

Dear Msc Urgesa,

Thank you very much for submitting your manuscript "Evidence for hidden leprosy in the post-elimination era in a high leprosy endemic setting, Eastern Ethiopia: application of active case finding and contact screening" for consideration at PLOS Neglected Tropical Diseases. As with all papers reviewed by the journal, your manuscript was reviewed by members of the editorial board and by several independent reviewers. In light of the reviews (below this email), we would like to invite the resubmission of a significantly-revised version that takes into account the reviewers' comments. 

Please pay special attention to recommendations regarding formatting, language, structure and length of the discussion, the concept of elimination of leprosy and limitations of the study.

We cannot make any decision about publication until we have seen the revised manuscript and your response to the reviewers' comments. Your revised manuscript is also likely to be sent to reviewers for further evaluation.

Sincerely,

Mauro Sanchez, ScD

Associate Editor

Gerson Penna

Deputy Editor

Reviewer's Responses to Questions

**Key Review Criteria Required for Acceptance?**

**Methods**

-Are the objectives of the study clearly articulated with a clear testable hypothesis stated?

-Is the study design appropriate to address the stated objectives?

-Is the population clearly described and appropriate for the hypothesis being tested?

-Is the sample size sufficient to ensure adequate power to address the hypothesis being tested?

-Were correct statistical analysis used to support conclusions?

-Are there concerns about ethical or regulatory requirements being met?

Reviewer #1: This is a descriptive study and investigatory of nature. Although quite a large population (app. 16,000) was screened for signs and symptoms of leprosy, in the end only 15 (hidden) cases were confirmed. This is important information from a policy point of view, but the low numbers make it difficult to perform statistical analyses satisfactorily. In fact, there are no statistically significant differences shown in any of the analyses at the p=0.05 level.

Reviewer #2: This study is observational epidemiologic description

Reviewer #3: See attached document

Reviewer #4: Yes. I just recommend inserting a map to locate the location of kebeles on the Ethiopian map for the international reader.

In the future, I recommend doing a case-control study (with neighborhood controls) to better estimate the contribution of these associated factors to the occurrence of the disease in this area.

**Results**

-Does the analysis presented match the analysis plan?

-Are the results clearly and completely presented?

-Are the figures (Tables, Images) of sufficient quality for clarity?

Reviewer #1: Table 1 is difficult to read. Firstly, numbers (N) and (%) are given in the same column, these should be in separate columns. Secondly, 1 decimal for percentages is sufficient. Thirdly, total N should show 100% in each row every time. Or else put the total first in a column and give the total of each category in a row underneath. So:

Sex Negative Positive

Male 145 (55.3%) 134 (92.4%) 11 (7.6%)

Female 177 (44.7%)

Total 262 (100%)

Age category (in years)

<15

etc.

Information on the part on the hidden prevalence is divided between text (end page 12 and page 13) and Figure 1. This is confusing. The figure does not add much extra information at all and this part can be best summarized in a table, bringing all information comprehensively together.

Table 2 is not necessary. The 4 lines (250-253) on page 13 say it all...

Reviewer #2: well presented and discussed

Reviewer #3: See attached document

Reviewer #4: - For the level of education only have these two possibilities? Wouldn't it be interesting to include, classify "literates" in the education levels, covering other school grades, including people who have completed regular education or higher education?

- When discussing the results, I consider it delicate to make this comparison with studies carried out in other countries with a high leprosy burden. And even if not all of the cases were multibacillary, it was a high percentage in the study in question (lines 306 - 312). 

 - Regarding the information on lines 334 and 335, was it really a failure of the HEWs to identify these co-prevalent cases or because these contacts were not at home at the time of the first visit?

**Conclusions**

-Are the conclusions supported by the data presented?

-Are the limitations of analysis clearly described?

-Do the authors discuss how these data can be helpful to advance our understanding of the topic under study?

-Is public health relevance addressed?

Reviewer #1: The discussion section is very lengthy and can be shortened by at least a third. The strengths and limitations usually come before the final conclusion. Most part of the conclusion paragraph are actually recommendations and could be formulated separately under a heading recommendations.

Reviewer #2: relevant and in accordance to tile and abstract

Reviewer #3: See attached document

Reviewer #4: - In lines 200 and 201 it was mentioned that "While individual confirmed as leprosy case initiated treatment, suspects with other skin diseases were linked to the nearby health facility for clinical management." On lines 275 and 276 "Another challenge in this study was insufficient or shortage of MDT for leprosy treatment. "How was this important ethical issue and articulation with health services resolved? Were all the cases diagnosed in the research treated as mentioned in the methodology? consent that participants signed? Make it clearer on the paper.

 - Lines 358 e 359 - I think that this discussion can be further developed in the light of the current literature on the social determinants of the occurrence of leprosy and considering the limitations of the study and data analysis

 - I suggest adding recommendations for other research that can complement the identified results.

**Editorial and Data Presentation Modifications?**

Reviewer #1: Indicate once in the text that the term for village in Ethiopia is kebele, but otherwise use the familiar English term village throughout the manuscript.

Reviewer #2: yes

Reviewer #3: (No Response)

Reviewer #4: - Attention to the formatting details of paragraphs, tables and graph.

**Summary and General Comments**

Reviewer #1: Although this paper is relevant for Ethiopia from a leprosy control policy point of view, its scientific (epidemiological) merit is rather limited due to the low numbers and very descriptive nature. It would be better suited for a specific leprosy journal, such as Leprosy Review. In any case, the manuscript can be shortened and improved (see specific comments). Also, it needs a very thorough English language edit. This is a major limitation of the current manuscript.

Reviewer #2: clear

Reviewer #3: See attached document

Reviewer #4: This issue of elimination is controversial because this indicator is based on a prevalence calculated based on cases with an active record, under treatment - when in fact, case detection and hidden prevalence should be considered. In the introduction and discussion I recommend doing a more in-depth reflection on this "post-elimination" era and what it really takes and what it means to achieve these leprosy elimination goals. How to consider that a disease has been eliminated, as mentioned in lines 71 to 75, if the indicator that represents this elimination disregards the hidden prevalence? I recommend this among other references:Lockwood, DN; Shetty, V; Penna, GO (2014) Hazards of setting tar-gets to eliminate disease: lessons from the leprosy elimination cam-paign. BMJ (Clinical research ed), 348. g1136. ISSN 0959-8138 DOI:https://doi.org/10.1136/bmj.g1136

PLOS authors have the option to publish the peer review history of their article (what does this mean?). If published, this will include your full peer review and any attached files.

Reviewer #1: No

Reviewer #2: Yes: Carlos Franco-Paredes

Reviewer #3: No

Reviewer #4: No
---

## [Decision Letter · Decision Letter 1]

9 Jul 2021

Dear Dr Urgesa,

We are pleased to inform you that your manuscript 'Evidence for hidden leprosy in a high leprosy-endemic setting, Eastern Ethiopia: the application of active case-finding and contact screening' has been provisionally accepted for publication in PLOS Neglected Tropical Diseases.

Best regards,

Mauro Sanchez, ScD

Associate Editor

Gerson Penna

Deputy Editor

Reviewer's Responses to Questions

**Key Review Criteria Required for Acceptance?**

**Methods**

-Are the objectives of the study clearly articulated with a clear testable hypothesis stated?

-Is the study design appropriate to address the stated objectives?

-Is the population clearly described and appropriate for the hypothesis being tested?

-Is the sample size sufficient to ensure adequate power to address the hypothesis being tested?

-Were correct statistical analysis used to support conclusions?

-Are there concerns about ethical or regulatory requirements being met?

Reviewer #1: The authors have adjusted the methods section well according to reviewer suggestions.

Reviewer #3: (No Response)

**Results**

-Does the analysis presented match the analysis plan?

-Are the results clearly and completely presented?

-Are the figures (Tables, Images) of sufficient quality for clarity?

Reviewer #1: The presentation of results has improved significantly according to reviewer suggestions.

Reviewer #3: (No Response)

**Conclusions**

-Are the conclusions supported by the data presented?

-Are the limitations of analysis clearly described?

-Do the authors discuss how these data can be helpful to advance our understanding of the topic under study?

-Is public health relevance addressed?

Reviewer #1: The discussion and conclusions have improved markedly and are now concise and to the point.

Reviewer #3: (No Response)

**Editorial and Data Presentation Modifications?**

Reviewer #1: Accept

Reviewer #3: Line 190 "tenderness and the presence of acid-fast bacilli in a slit-skin smear" needs a little correction.

The presence of bacilli is not necessary to establish the diagnosis of leprosy. To better clarify, as a suggestion:

"with or without tenderness and with or without the presence of acid-fast bacilli in a slit-skin smear."

**Summary and General Comments**

Reviewer #1: I think that the revised manuscript now better highlights the fact that unexpected relatively high numbers of new cases are found through active case finding in endemic districts and its relevance for leprosy control measures. There is now less (unnecessary) emphasis on statistical analyses, which distracted from the main message of the paper.

Reviewer #3: Congratulations on the excellent work.

PLOS authors have the option to publish the peer review history of their article (what does this mean?). If published, this will include your full peer review and any attached files.

Reviewer #1: No

Reviewer #3: No

---

## [Editor Report · Acceptance letter]

4 Aug 2021

Dear Mr URGESA,

We are delighted to inform you that your manuscript, "Evidence for hidden leprosy in a high leprosy-endemic setting, Eastern Ethiopia: the application of active case-finding and contact screening," has been formally accepted for publication in PLOS Neglected Tropical Diseases.

Best regards,

Shaden Kamhawi

co-Editor-in-Chief

Paul Brindley

co-Editor-in-Chief
